# Quantitative Analysis of Effects of a Single ^60^Co Gamma Ray Point Exposure on Time-Dependent Change in Locomotor Activity in Rats

**DOI:** 10.3390/ijerph17165638

**Published:** 2020-08-05

**Authors:** Keiko Otani, Megu Ohtaki, Nariaki Fujimoto, Aisulu Saimova, Nailya Chaizhunusova, Tolebay Rakhypbekov, Hitoshi Sato, Noriyuki Kawano, Masaharu Hoshi

**Affiliations:** 1The Center for Peace, Hiroshima University, Hiroshima 730-0053, Japan; ohtaki@hiroshima-u.ac.jp (M.O.); nkawano@hiroshima-u.ac.jp (N.K.); mhoshi@hiroshima-u.ac.jp (M.H.); 2Research Institute for Radiation Biology and Medicine, Hiroshima University, Hiroshima 734-8553, Japan; nfjm@hiroshima-u.ac.jp; 3Maxillo-facial and facial plastic surgery, Semey Medical University, Semey 071400, Kazakhstan; aisulu626@gmail.com; 4Public health, Medical University, Semey 071400, Kazakhstan; n.nailya@mail.ru; 5Board of Directors, Astana Medical University, Astana 010000, Kazakhstan; tolebay13@gmail.com; 6Radiological Sciences, Ibaraki Prefectural University of Health Sciences, Ibaraki 300-0394, Japan; satoh@ipu.ac.jp

**Keywords:** external irradiation, locomotor activity, individual differences, non-linear mixed effects model, time-dependency

## Abstract

Investigating initial behavioral changes caused by irradiation of animals might provide important information to aid understanding of early health effects of radiation exposure and clinical features of radiation injury. Although previous studies in rodents suggested that radiation exposure leads to reduced activity, detailed properties of the effects were unrevealed due to a lack of proper statistical analysis, which is needed to better elucidate details of changes in locomotor activity. Ten-week-old male Wistar rats were subjected to single point external whole-body irradiation with ^60^Co gamma rays at 0, 2.0, 3.5, and 5.0 Gy (four rats per group). Infrared sensors were used to continuously record the locomotor activity of each rat. The cumulative number of movements during the night was defined as “activity” for each day. A non-linear mixed effects model accounting for individual differences and daily fluctuation of activity was applied to analyze the rats’ longitudinal locomotor data. Our statistical method revealed characteristics of the changes in locomotor activity after radiation exposure, showing that (1) reduction in activity occurred immediately—and in a dose-dependent manner—after irradiation and (2) recovery to pre-irradiation levels required almost one week, with the same recovery rate in each dose group.

## 1. Introduction

In humans, one of the earliest effects of radiation exposure to the whole body or to a large portion of the whole body is a prodromal period of nonspecific signs and symptoms such as nausea, emesis, fatigue, fever, and anorexia [1,2]. The prodromal syndrome is generally mild or absent at total body doses of 1 Gy or less and occurs from minutes to days following exposure [3,4,5]. However, it is unclear to what extent these symptoms are psychogenic versus radiation-induced. Therefore, the relationship between initial symptoms and radiation dose is not well understood. 

Early effects of irradiation have been studied in regard to radiation therapy. In a detailed study of the incidence and severity of side effects during the course of radiation therapy, fatigue was the most prevalent and the most severe symptom reported by patients [6]. With fractionated doses of radiation for cancer treatment, radiation-induced fatigue sets in within a few days after start of treatment and decreases after treatment completion [7]. Although the underlying mechanisms of fatigue have been studied under several disease conditions, an understanding of the etiology, mechanisms, and risk factors of radiation-induced fatigue remains elusive, and this symptom remains poorly managed [8,9,10]. Investigating initial radiation-related behavioral changes by using animals might provide important information to aid understanding of the health effects of radiation exposure and clinical features of radiation injury.

In animals, there have been many studies of radiation-induced behavioral effects, and performance decrement after irradiation has been noted in several reports. A sub-lethal dose of gamma radiation suppressed aggressive behavior in male mice [11], a lethal dose of gamma radiation suppressed locomotor activity in mice [12], and a sub-lethal dose of X-irradiation suppressed volitional activity in rats [13]. Landauer (2002) provided a review of expected performance decrement after radiation exposure [14]. These reports showed that ionizing radiation temporarily suppresses animals’ behavior, but that the effect does not continue for a long period. York et al. reported that, 6 h after gamma irradiation with 50 or 200 cGy, spontaneous locomotor activity in mice was 35% or 36% lower, respectively, than in sham irradiated controls, and that their activity recovered to sham irradiated level 12 h after irradiation [15].

Although many animal behavioral experiments have a time-dependent data structure with variation among individuals, analyses have typically been performed only at individual time points with no parameterization of the trend in activity over time. Therefore, quantitative analyses have not been made directly on the chronological features. To obtain more detailed and accurate information from data obtained in animal behavior experiments with time-dependent structure and individual variability, application of statistical theory would suggest that analysis based on a mixed effects model [16,17] is both appropriate and effective.

The reason for the experiments was to investigate the early health effects of radiation exposure. Then, statistical models were used to examine in detail the changes over time in locomotor activity of rats immediately after external irradiation with ^60^Co gamma rays. Specifically, we aimed to assess the time when reduction of locomotor activity begins, the time when locomotor activity recovers to pre-irradiation level, the dose dependency of the degree of reduction in locomotor activity, and the dose dependency of the rate of recovery. There are individual differences in animal behavior that cannot be ignored, even if the animal type, gender, and weight are uniform. In addition, when animals are observed over a long period of time, it is expected that common changes in behavior will occur due to indoor conditions such as temperature, humidity, and noise, which can change daily, and it is necessary to adjust for these sources of variation.

## 2. Materials and Methods 

### 2.1. Experimental Design and Data Collection

#### 2.1.1. Animals 

The experiment was approved by the Animal Experiment Committee of Semey Medical University, Republic of Kazakhstan, and was conducted in accordance with the Institutional Guide for Animal Care and Use. Ten one-week-old male Wistar rats were purchased from the Kazakh Scientific Center of Quarantine and Zoonotic Diseases, Almaty, Kazakhstan and allowed free access to a basal diet and tap water. Animal rooms were maintained at 19–22 °C with relative humidity 30–70% and a 12 h light cycle. Body weights were measured twice a week during the experiment. At 11 weeks of age, the rats were randomly divided into four groups: control (4 rats) and three irradiated groups (4 rats/group). Each irradiated group received 2, 3.5, or 5.0 Gy of whole body gamma irradiation. Controls were handled with all conditions the same as with the other groups, except that they were not irradiated (dose 0 Gy). The LD_50(30)_ for this strain of Wistar rats is 7 Gy with cobalt-60 radiation [18]. 

#### 2.1.2. Irradiation with ^60^Co Gamma-Rays 

Irradiation was performed with a Teragam K-2 unit (UJP Praha, Praha-Zbraslav, Czech Republic) at the Regional Oncology Dispensary of Semey. Rats were irradiated at 1 m distance from the ^60^Co source at a dose rate of 2.6 Gy/min. Half of the radiation dose was administered from the top and the other half was administered from the bottom. A radiophotoluminescence glass dosimeter, GD-302M [Chiyoda Technol Co., Tokyo, Japan], was used for measuring the doses. 

#### 2.1.3. Measurements of Daily Locomotor Activity

Locomotor activities (hereafter abbreviated as “activities”) of the rats were measured with infra-red sensors (Model NS-AS01; Neuroscience, Inc., Tokyo, Japan) placed 16 cm above the open-top cages (26.5 × 43 × 14.5 cm). Numbers of movements were counted on the basis of change in the strength of infra-red rays emitted from the animals. The rats were placed in separate cages, each outfitted with a sensor, and movements were continuously counted by a computerized analysis system (16 channel Multi-digital Counter System [MDC] and DAS System software, Neuroscience, Inc. Tokyo, Japan). Measurements were started 3 days before irradiation and continued for 20 days after irradiation. 

#### 2.1.4. Ethical Approval

All applicable international, national, and/or institutional guidelines for the care and use of animals were followed. The animal experiment was approved by the Animal Experiment Committee of Semey Medical University, Republic of Kazakhstan (Protocol No 5 dated 16.04.2014), and conducted in accordance with the Institutional Guide for Animal Care and Use.

### 2.2. Statistical Analyses

#### 2.2.1. Definition of Daily Activity

Because rats are nocturnal animals [19], the cumulative number of movements was recorded during the period between 18:00 and 06:00; the number of movements so recorded was defined as activity of a rat in one day. As shown in Figure 1, rates of increase in cumulative movements (slopes) were steeper during nighttime (18:00–05:59) than during daytime (06:00–17:59); i.e., the rats were more active at night, as expected. 

This suggests that the activity defined in this study represents the nocturnal characteristic of rats and it shows that the measure has relevance as an indicator of a rat’s activity.

#### 2.2.2. Data Modeling

Logarithmic values of daily activity of each rat as a function of elapsed time relative to day of irradiation are shown for each group in Figure 2. 

An acute decrease in activity after irradiation followed by quick recovery to the pre-irradiation level can be seen in every exposed group, whereas no such change or trend was observed in the control group. There was also large inter-animal variation with daily fluctuation in activity. Therefore, we assumed a non-linear mixed effects model (NLMM) [16,17] that takes into account the dose dependency of the decrease in activity, the dose dependency of the recovery rate, individual differences among animals, and daily fluctuations within individual animals. For comparison, we fit a simple non-linear regression model (NLRM) in which individual differences and daily fluctuations were not taken into account.

#### 2.2.3. Non-Linear Mixed Effects Model (NLMM)

Let $y_{it}$ be the log transformed observed activity of rat i at time t in days since irradiation with dose $D_{i}\;(t=-3,\ldots ,20\;;\;i=1,\ldots ,16)$, where “t=0” indicates day of irradiation. We assume the model:$$y_{it}=f(t|D_{i},\theta )+\delta_{i}+\eta_{t}+\varepsilon_{i\;t}$$
$$f(t|D_{i},\;\theta )=\xi_{0}+\xi_{1}t+\xi_{2}t^{2}-\left(\beta_{1}D_{i}+\beta_{2}D_{i}{}^{2}\right)\cdot \exp\left[\left\{-\omega_{1}\cdot e^{-\omega_{2}(D_{i}-D_{0})}\right\}t\right]\cdot h(t)$$
(1)$$\delta_{i}∼N(0,\;\psi^{2}),\;\eta_{t}∼N(0,\;\varphi^{2}),\;\varepsilon_{it}∼N(0,\;\sigma^{2}),t=-3,-2,\ldots ,20,\;i=1,\ldots ,\;16,\;$$
where $\theta =(\xi_{0},\;\xi_{1}\;,\;\xi_{2},\;\beta_{1},\;\beta_{2},\;\omega_{1},\omega_{2})$ denotes unknown parameters for fixed effects to be estimated. 

The term $\xi_{0}+\xi_{1}t+\xi_{2}t^{2}$ expresses the time dependency of activities without radiation exposure. The term $\beta_{1}D_{i}+\beta_{2}D_{i}{}^{2}$ expresses whether the dose effect in the initial decrease is linear ($\beta_{2}=0$) or quadratic ($\beta_{2}\neq 0$), and the term $-\omega_{1}\cdot e^{-\omega_{2}(D_{i}-D_{0})}$ denotes whether the recovery rate depends on dose ($\omega_{2}\neq 0$) or not ($\omega_{2}=0$). $D_{0}$ denotes a fixed pre-assigned dose value for covariate centering (in this study 2.75 Gy is adopted), $\;\Delta =(\psi^{2},\;\varphi^{2},\sigma^{2})$ are unknown dispersion parameters to be estimated, and the terms $\delta_{i},\;\eta_{t}$, and $\varepsilon_{it}$ represent independent random effects due to individual variability, daily fluctuation, and measurement error, respectively. The function  h(t):h(t)=0 (t<0),  h(t)=1 (t≥0) denotes the Heaviside function of t to indicate pre- and post-irradiation dichotomy. 

Let $y=({y_{1}}^{\prime },\ldots ,{y_{16}}^{\prime }){}^{\prime },\;y_{i}=(y_{i,-3},\ldots ,y_{i,20}){}^{\prime },\;i=1,\ldots ,16$. It follows from Model (1) that y has a multivariate normal distribution with mean $\mu (\theta )=(\mu_{1}(\theta ){}^{\prime },\ldots ,\mu_{16}(\theta ){}^{\prime }){}^{\prime },\;\mu_{i}(\theta )=f(t|D_{i},\theta ),\;t={\left(-3,-2,\ldots ,20\right)}^{\prime },\;i=1,\ldots ,16,$ and variance-covariance matrix $\Omega (\Delta )=I_{16}⊗\left(\rho^{2}J_{41}+\sigma^{2}I_{41}\right)+J_{16}⊗\psi^{2}I_{41},$ where $I_{m}$ denotes an m-dimensional unit matrix, and $J_{m}=1_{m}⊗{1_{m}}^{\prime }.$ Then the likelihood function of (θ,Δ) can be expressed as $L(\theta ,\Delta )=\frac{1}{{\left(2\pi \right)}^{8}\sqrt{\left|\Omega (\Delta )\right|}}\exp\left(-\frac{1}{2}{\left\{y-\mu (\theta )\right\}}^{\prime }\Omega {\left(\Delta \right)}^{-1}\left\{y-\mu (\theta )\right\}\right).$ Therefore, the maximum likelihood estimates of (θ,Δ), denoted by $\left(\hat{\theta },\hat{\Delta }\right)$, are obtained by minimizing the quantity $Q(\theta ,\Delta )=\log\left(\left|\Omega (\Delta )\right|\right)+{\left\{y-\mu (\theta )\right\}}^{\prime }\Omega {\left(\Delta \right)}^{-1}\left\{y-\mu (\theta )\right\}+16\times \log(2\pi )$. When $\psi^{2}=\varphi^{2}=0$, Model (1) reduces to an ordinary non-linear regression model (NLRM). 

#### 2.2.4. Algorithm and Software for Implementation of Data Analyses 

The unknown parameters were estimated by using an algorithm for optimization with the limited-memory version of the Broyden–Fletcher–Goldfarb–Shanno method [20] to maximize the likelihood derived from the model (1), and the AIC (Akaike Information Criterion) [21] and BIC (Bayesian information criterion) [22,23] were calculated. The function ‘optim’ in the R software ver. 3.5.1 was used for carrying out numerical analyses. 

Maximum likelihood (ML) or restricted maximum likelihood (REML) [24] estimates of the parameters in the linear mixed-effects models can be computed with the “lmer” function in the “lme4” package for R [25]. In this study, the ML method was used to compare the goodness-of-fit of models with the AIC criterion. Estimation results were almost the same with both methods. 

## 3. Results

### 3.1. Result of Regression Analysis

#### 3.1.1. Estimation of Fixed Effect Parameters

Regression analysis was first performed with all parameters of the NLMM (full NLMM), then model selection was applied by choosing the smallest AIC to determine the optimal NLMM (optimal NLMM). The full NLRM and optimal NLRM were defined in the same way. Estimates of fixed-effect parameters and their 95% confidence intervals under the full and optimal NLMM are shown in Table 1a and Table 1b respectively; those under the full and optimal NLRM are shown in Table 2a and Table 2b, respectively.

#### 3.1.2. Estimation of the Random Effects Parameters

In the optimal NLMM, variances of the random effects due to individual differences, daily variation, and measurement error were 0.0018, 0.0019, and 0.0015, which account for 35%, 36%, and 29% of the total variance, respectively. Predictions of individual differences $\left({\hat{\delta }}_{1},{\hat{\delta }}_{2},\cdots ,{\hat{\delta }}_{16}\right)$ and those of daily fluctuation $\left({\hat{\eta }}_{-3},\;{\hat{\eta }}_{-2},\cdots ,{\hat{\eta }}_{20}\right)$ were obtained by calculating posterior means. The predictions ${\hat{\delta }}_{i}$ in each of the four groups (control group and three irradiated groups) and the predictions ${\hat{\eta }}_{t}$ by day are shown in panels (a) and (b) of Figure 3, respectively. 

Residuals in the optimal NLMM and in the optimal NLRM are given by $\;y_{it}-\hat{f}(t|D_{i},\theta )-{\hat{\delta }}_{i}-{\hat{\eta }}_{t}$ and $y_{it}-\hat{f}(t|D_{i},\theta )\;$, respectively. The standard deviations of residual errors in the optimal NLMM and optimal NLRM were 0.038 and 0.071, respectively. The distributions of residuals in the NLMM and NLRM are shown in Figure 4. 

### 3.2. Comparison of Goodness of Fit of the NLMM and the NLRM

There is a large difference between the AICs of the optimal NLMM and the optimal NLRM, which were −1271.80 and −930.25, respectively (See Table 1b and Table 2b). The measurement error variances of the NLMM and NLRM were 0.0015 and 0.0058 (See Table 1b and Table 2b). Therefore the fit of the NLMM was preferable to that of the NLRM in terms of prediction and accuracy. The estimated time dependency of activity in each group under the optimal NLMM is shown in Figure 5.

In each of the irradiated groups, activity decreased immediately after irradiation but recovered to the pre-irradiation level within a few days with a common recovery rate irrespective of dose. 

## 4. Discussion

One of the advantages of using the more complex NLMM structure, as demonstrated in this paper, is that a second-order dose dependency could be detected in the initial decrease, which was not found with the NLRM (which estimated a linear dependency). Estimated magnitudes of initial decreases at t=0 by dose group and their 95% confidence intervals in the optimal NLMM and those in the optimal NLRM are shown in Figure 6. 

The plots of predictions of individual differences ${\hat{\delta }}_{i}$ by dose group (Figure 3a) show that the assumption of homoscedasticity for distributions of individual difference between the four dose groups seems to be satisfied. This means that the random assignment of rats to the four groups was effective in terms of individual differences. The plots of predictions of time-dependent daily fluctuation ${\hat{\eta }}_{t}$ (Figure 3b) show that the assumption of independency of each of the random variables $\eta_{t}$ seems to be satisfied. The Durbin Watson statistic [26] for ${\hat{\eta }}_{t}$ was 2.33 (*p*-value 0.902), indicating that no strong autocorrelation is observed in daily fluctuation. 

Because acute changes were the focus in this experiment, longer observation was not performed, but it is necessary to investigate late effects. The irradiation was a single and sub-lethal dose, so it is considered that damage was acute, disappearing in a short period of time, and resilience to allow recovery from the damage was not affected by irradiation. The effects of chronic low dose exposure remain as future issues to be addressed. As one important example of the need for assessing effects of chronic exposure, a giant earthquake of magnitude M9 struck East Japan on 11 March 2011. Subsequently a ‘tsunami’ engulfed the Fukushima Daiichi Nuclear Power Plant (FDNPP). As a result, FDNPP reactors 1−3 suffered meltdown and significant amounts of radioactive materials were released into the environment [27]. The dose to the public is estimated to be low [28], but many Japanese people are worried about the resulting health effects of chronic low dose exposure. 

## 5. Conclusions

In the present study, the effects of irradiation on the behavior of rats were investigated efficiently, despite a small number of animals with large individual differences. This was achieved by using a statistical method that accounts for inter-animal differences and daily fluctuation in activity—a non-linear mixed model fit to repeated measurements. With such an efficient approach, we were able to demonstrate a temporary, but dose-dependent, decrease in activity following irradiation and a dose-independent common recovery rate. The statistical framework for analyzing longitudinal locomotor data in this study should be generally applicable to other repeated measurement data with a similar structure. 

## Figures and Tables

**Figure 1 ijerph-17-05638-f001:**
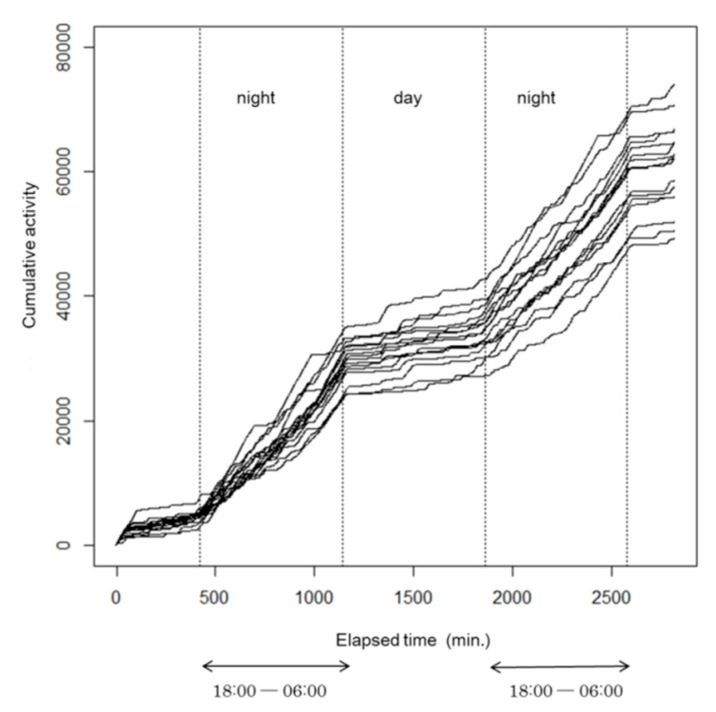
Cumulative number of movements of each of the 16 rats over a 36-h period.

**Figure 2 ijerph-17-05638-f002:**
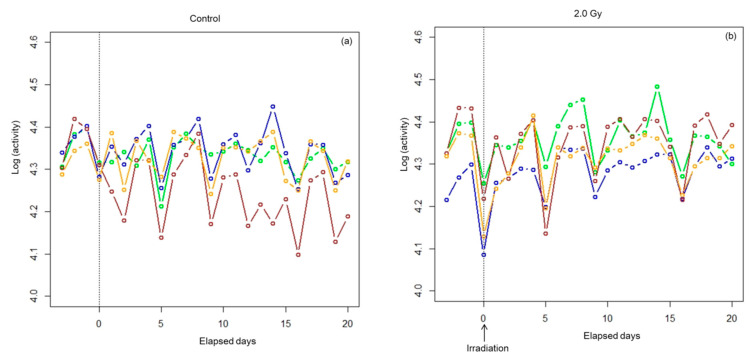
Daily activity of each of the four rats belonging to the four groups. The vertical axis shows the logarithm of daily activity (number of nocturnal movements) and the horizontal axis shows elapsed time in days relative to the day of irradiation (indicated by arrows): (**a**) the control group, (**b**) 2.0 Gy group, (**c**) 3.5 Gy group, and (**d**) 5.0 Gy group.

**Figure 3 ijerph-17-05638-f003:**
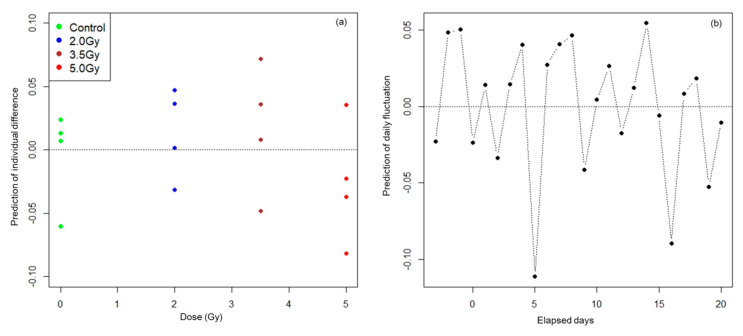
Predictions of random values. Predictions of random values by individual ${\hat{\delta }}_{i}$ by group are shown in panel (**a**) and predictions of random values by day ${\hat{\eta }}_{t}$ are shown in panel (**b**).

**Figure 4 ijerph-17-05638-f004:**
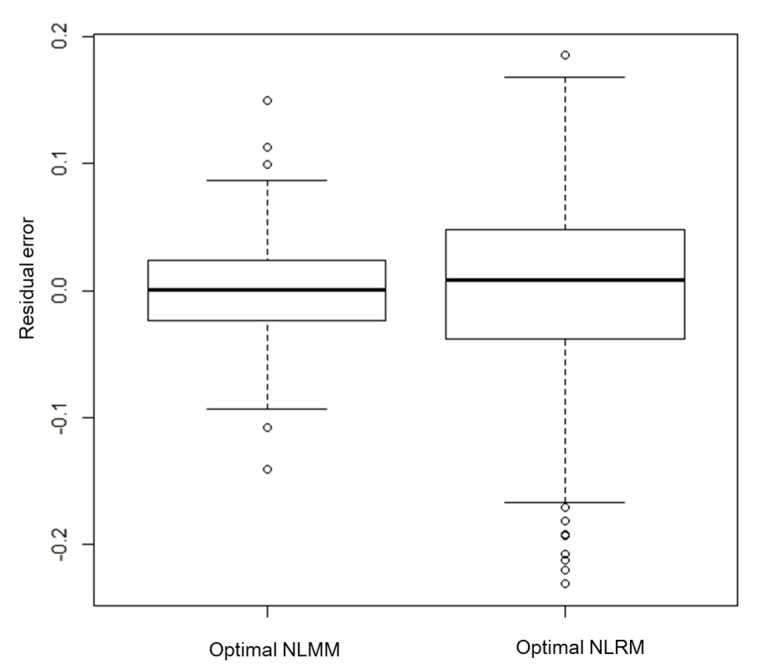
Parallel boxplots of residual errors in the non-linear mixed model (NLMM) and ordinary non-linear regression model (NLRM).

**Figure 5 ijerph-17-05638-f005:**
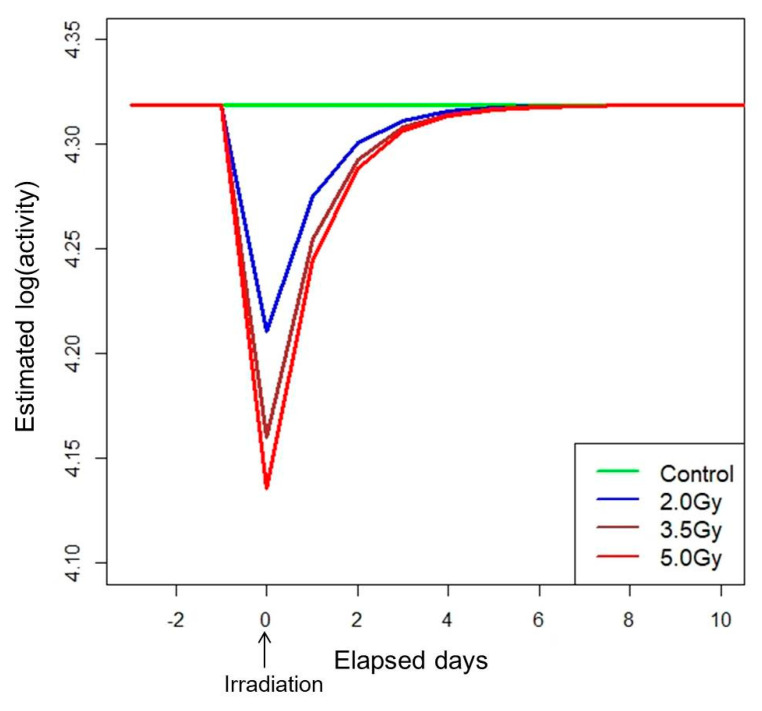
Estimated mean trends of daily locomotor activity in rats by dose group under the optimal NLMM.

**Figure 6 ijerph-17-05638-f006:**
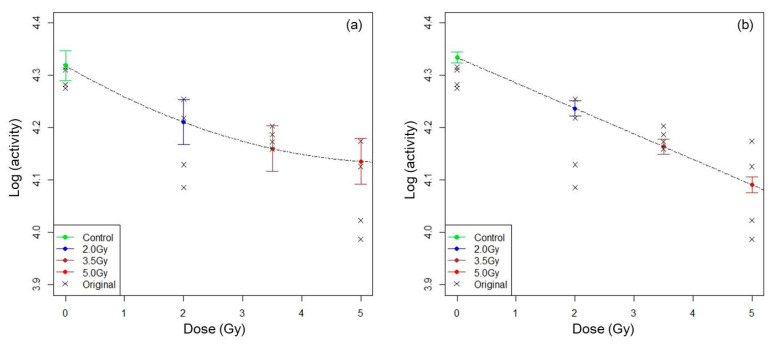
Fitted dose-response curves from the optimal NLMM and the optimal NLRM. The estimated magnitudes of decrease at t=0 by dose group and their 95% confidence intervals and fitted dose-response curves with dotted line from the NLMM and the NLRM are shown in panels (**a**) and (**b**), respectively. Cross marks show observed data of individual rats. The fitted dose-response curve from the optimal NLMM was a downward convex quadratic curve.

**Table ijerph-17-05638-t001a:** 

(a)	Full NLMM				
Parameter	Estimate	SE	95% Confidence Interval		*p*-Value
			Lower Bound	Upper Bound	
β_1_	0.069	0.015	0.041	0.098	0.000 **
β_2_	−0.007	0.003	−0.012	−0.001	0.023 *
ω_1_	10.391	4.808	0.968	19.815	0.015 *
ω_2_	0.082	0.166	−0.243	0.407	0.310
ξ_0_	4.326	0.020	4.288	4.364	0.000 **
ξ_1_	0.003	0.041	−0.076	0.083	0.468
ξ_2_	−0.008	0.022	−0.052	0.035	0.353

(a): **: p<0.01, *: 0.01≤p<0.05. Estimated random effect parameters: $\left(\hat{\psi^{2}},\;\hat{\varphi^{2}},\;\hat{\sigma^{2}})=(0.0018,\;0.0019,\;0.0015\right)$. Log-likelihood: 643.47, AIC: −1266.94, BIC: −1227.44.

**Table ijerph-17-05638-t001b:** 

(b)	Optimal NLMM				
Parameter	Estimate	SE	95% Confidence Interval		*p*-Value
			Lower Bound	Upper Bound	
β_1_	0.066	0.016	0.033	0.098	0.000 **
β_2_	−0.006	0.003	−0.012	0.001	0.036 *
ω_1_	9.063	2.949	3.283	14.843	0.001 **
ξ_0_	4.319	0.014	4.290	4.347	0.000 **

(b): **: p<0.01, *: 0.01≤p<0.05. Estimated random effect parameters: $\left(\hat{\psi^{2}},\;\hat{\varphi^{2}},\;\hat{\sigma^{2}})=(0.0018,\;0.0019,\;0.0015\right)$. Log-likelihood: 642.90, AIC: −1271.80, BIC: −1244.15.

**Table ijerph-17-05638-t002a:** 

(a)	Full NLRM				
Parameter	Estimate	SE	95% Confidence Interval		*p*-Value
			Lower Bound	Upper Bound	
β_1_	0.075	0.023	0.030	0.120	0.001 **
β_2_	−0.006	0.005	−0.016	0.004	0.104
ω_1_	3.922	1.404	1.170	6.674	0.003 **
ω_2_	0.574	0.447	−0.303	1.450	0.100
ξ_0_	4.333	0.007	4.320	4.346	0.000 **
ξ_1_	0.003	0.016	−0.028	0.033	0.435
ξ_2_	−0.011	0.008	−0.027	0.006	0.107

(a): **: p<0.01, *: 0.01≤p<0.05. Estimated residual variance: $\hat{\sigma^{2}}=0.00502$. Log-likelihood: 744.091, AIC: −928.17, BIC: −883.56.

**Table ijerph-17-05638-t002b:** 

(b)	Optimal NLRM				
Parameter	Estimate	SE	95% Confidence Interval		*p*-Value
			Lower Bound	Upper Bound	
β_1_	0.049	0.005	0.039	0.059	0.000 **
ω_1_	5.973	1.726	2.590	9.356	0.000 **
ξ_0_	4.334	0.006	4.323	4.345	0.002 **
ξ_2_	−0.010	0.003	−0.016	−0.004	0.000 **

(b): **: p<0.01, *: 0.01≤p<0.05. Estimated residual variance: $\hat{\sigma^{2}}=0.0058$. Log-likelihood: 742.12, AIC: −930.25, BIC: −899.02.

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
