# Peer review of "Quantitative Analysis of Effects of a Single 60Co Gamma Ray Point Exposure on Time-Dependent Change in Locomotor Activity in Rats"

_ijerph, 2020, doi:10.3390/ijerph17165638_

Round 1
Reviewer 1 Report
The paper describe a statistical method to detect a damage from irradiation . the approach is interesting however some physiological parameters could be introduced and compared with statistical model. Than the statistical tool alone cannot used to detect damages or not.
Moreover is not very clear the aim of that, so in the introduction must be explained the reasons for the irradiation. Than the statistical tool alone cannot used to detect damages or not.
Author Response
Point 1: Some physiological parameters could be introduced and compared with statistical model. Than the statistical tool alone cannot used to detect damages or not.
Response 1:
As Reviewer 1 points out, radiation damage can have a variety of physiological consequences such as a decrease in white blood cell count. In fact, this experiment also observed weight loss after irradiation.
The reason for carrying out the experiments was to investigate the early health effects of radiation and the purpose of this study was to clarify the quantitative effects of radiation damage on behavioral changes in animals. Statistical modeling is not a priority over experimental research. According to Reviewer 1's suggestion, the introduction was modified by adding a description to the purpose of the experiment for clarity this.
Point 2: Moreover is not very clear the aim of that, so in the introduction must be explained the reasons for the irradiation. Than the statistical tool alone cannot used to detect damages or not.
Response 2:
We rewrite the first sentence in the last paragraph of the Introduction (line 77-79) as follows: The reason for the experiments was to investigate the early health effects of radiation exposure. Then, statistical models were used to examine in detail the changes over time in locomotor activity of rats immediately after external irradiation with 60Co gamma rays.
Reviewer 2 Report
This was just a small commentary on this manuscript. From a statistical point of view, it is best if a very large sample is used in the study. In studies in which animals are used, the statistical requirements and the number of animals used must be appropriately centered. I stand by my decision to accept this work and I would not suggest any changes to the authors.
Author Response
Point: From a statistical point of view, it is best if a very large sample is used in the study. In studies in which animals are used, the statistical requirements and the number of animals used must be appropriately centered.
Response:
It was difficult to prepare a lot of samples, because the experimental equipment was expensive. If we had an additional data at low dose level, for example at 1 Gy, we might have been able to obtain more detail results.
Round 2
Reviewer 1 Report
I accept revisions